# Microglia and p38 MAPK Inhibitors Suppress Development of Mechanical Allodynia in Both Sexes in a Mouse Model of Antiretroviral-Induced Neuropathic Pain

**DOI:** 10.3390/ijms241612805

**Published:** 2023-08-15

**Authors:** Maryam W. Alhadlaq, Willias Masocha

**Affiliations:** 1Molecular Biology Program, College of Graduate Studies, Kuwait University, Safat, Kuwait City 13110, Kuwait; maryam.alhadlaq@grad.ku.edu.kw; 2Department of Pharmacology and Therapeutics, College of Pharmacy, Kuwait University, Safat, Kuwait City 13110, Kuwait

**Keywords:** antiretroviral drug, ddC, minocycline, adezmapimod, neuroinflammation, neuropathic pain, mechanical allodynia, microglia, sex differences

## Abstract

Microglia activation in the spinal cord play a major role in the pathogenesis of neuropathic pain. The p38 mitogen-activated protein kinase (MAPK) regulates microglia activation. Previously, 2′,3′-dideoxycytidine (ddC), a nucleoside reverse transcriptase inhibitor (NRTI), was found to induce mechanical allodynia and microglia activation in the spinal cords of male and female mice. In this study, we investigated the role of spinal microglia and p38 MAPK signaling in the development of mechanical allodynia using immunofluorescence staining and treatment with microglia and p38 MAPK inhibitors in both sexes. Male and female mice (BALB/c strain) treated intraperitoneally once daily with ddC 25 mg/kg for five consecutive days developed mechanical allodynia, assessed using the dynamic plantar aesthesiometer. Treatment with ddC increased microglia markers CD11b and ionized calcium-binding adapter molecule 1 (Iba1) staining intensity in male mice, while only CD11b was increased in female mice. Both sexes had increased phosphorylated p38 MAPK staining intensity. The administration of minocycline, an inhibitor of microglia activation, and adezmapimod, a selective p38 MAPK inhibitor, suppressed mechanical allodynia in both sexes at day 7 after ddC treatment. Therefore, microglia activation and p38 MAPK signaling are important for the development of antiretroviral drug-induced mechanical allodynia.

## 1. Introduction

Neuropathic pain is a chronic pain that is caused or initiated by primary lesions or dysfunction in the central nervous system (CNS) or the peripheral nervous system (PNS) [1] and characterized by spontaneous and/or evoked pain in response to noxious (hyperalgesia) or non-noxious (allodynia) stimuli [2]. It is a serious clinical problem that affects millions of people worldwide, with a prevalence ranging between 3% and 17% [3]. A significant portion of people living with human immunodeficiency virus (HIV) suffer from peripheral neuropathies, which substantially affect their quality of life [4]. Antiretroviral therapy (ART) uses a combination of drugs to manage HIV. Nucleoside reverse transcriptase inhibitors (NRTIs) are the cornerstone antiretroviral drugs utilized in combination therapy used to treat HIV patients [5]. However, some NRTIs, including stavudine (d4T), didanosine (ddI), and zalcitabine (2′,3′-dideoxycytidine, ddC), were found to cause peripheral neuropathy [6]. Recently, we found that ddC induced mechanical allodynia and neuroinflammation in the spinal cords of male and female mice [7], similar to what was reported in a previous study that studied and observed neuroinflammation in the brains of female mice [8].

In neuropathic pain models using rodents, it has been shown that the activation of microglia cells in the spinal cord contribute to induce neuroinflammation and neuropathic pain [7,8,9,10,11,12,13]; thus, inhibiting the activation of microglia might be a possible approach for preventing neuropathic pain. The administration of minocycline, an inhibitor of microglia activation, attenuated mechanical allodynia in different models of neuropathic pain, including female mice with ddC-induced mechanical allodynia [8,14,15,16,17,18]. Preclinical studies revealed that the activation of p38 MAPK in the microglia cells of the spinal cord plays a role in initiating neuropathic pain [19,20,21], resulting in an increased release of pro-inflammatory cytokines, such as tumor necrosis factor-alpha (TNF-α) and interleukin 1 (IL-1β) [22,23]. The inhibition of p38 MAPK has been shown to effectively reverse mechanical allodynia and reduce pain in various models of neuropathic pain [19,20,24,25]. The efficacy of minocycline and p38 MAPK inhibitors could be different in other models of neuropathic pain. For example, in a chronic construction injury (CCI) model, intrathecally administered minocycline and p38 MAPK inhibitors were effective in alleviating allodynia in male mice but not in female mice [26], while intraperitoneally administered minocycline prevented the development of mechanical allodynia in female mice with NRTI-induced neuropathic pain [8].

The majority of preclinical research in pain used only male rodents to avoid females’ hormone cycle, which might complicate the results [27]; yet, it is critical to consider sex as a variable. Evidence demonstrates that women represent the majority of patients with chronic pain, especially neuropathic pain [28,29]. Several studies revealed significant sex differences in neuroimmune responses and pain [25,26,30,31,32]. In a previous study, we reported that ddC administration induced mechanical allodynia equally in both male and female mice. However, using the whole spinal cord and Western blot analysis, microglia were activated in both male and female mice, but phosphorylated p38 MAPK was upregulated significantly only in female mice [7]. Thus, the objectives of this study were to evaluate, using immunofluorescence staining of the lumbar segment of the spinal cord, if (i) there are sex-specific activation patterns of microglia and phospho-p38 MAPK expression and also whether (ii) minocycline or p38 MAPK inhibitor (adezmapimod) have antiallodynic effects in a mouse model of NRTI-induced neuropathic pain. In this study, ddC was used as an NRTI-induced neuropathic pain model because the recommended initial ART regimens for naïve patients with HIV involve a combination of drugs. Nucleoside reverse transcriptase inhibitors are considered cornerstone for ART because they are used in all the combination therapies used to treat HIV patients. HIV-associated neuropathic pain, including that caused by ART, affects millions of people living with AIDS and is difficult to treat. The response to treatment differs between different types of neuropathic pain. The other models of neuropathic pain, such as partial sciatic nerve transection, CCI, and sciatic nerve compression or ligation, are good models but do not serve as models for the pain caused by antiretroviral drugs.

## 2. Results

### 2.1. ddC Induced Microglia Reaction on the Lumbar Segment of the Spinal Cord of Both Male and Female Mice

We previously reported that ddC increased the protein expression of Iba1 in the spinal cords of male—but not female—mice, while ddC upregulated the gene and protein expression of CD11b in the spinal cords of female—but not male—mice [9]. To assess whether microglia cells were activated in both sexes and whether there were sex-dependent differences in microglia markers in the ddC-induced neuropathic pain, we performed CD11b and Iba1 immunofluorescence staining on the lumbar segment of the spinal cords of both male and female mice 7 days after ddC administration. We found a significant increase in CD11b and Iba1 staining intensity in ddC-treated male mice compared to control vehicle-treated male mice (*p* < 0.05; Figure 1). In contrast, ddC-treated female mice had similar levels in Iba1 staining intensity compared to control vehicle-treated female mice (*p* > 0.05; Figure 2F), but there was a significant upregulation in CD11b staining intensity in ddC-treated female mice compared to control vehicle-treated female mice (*p* < 0.05; Figure 2C). When male and female mice were compared, there were sex differences in CD11b and Iba1 staining intensity post-ddC treatment; i.e., CD11b was more in female than male mice, while Iba1 was more in male than female mice after ddC treatment (*p* < 0.05; Figure 3A and Figure 3B, respectively).

### 2.2. ddC Induced p38 MAPK Activation on the Lumbar Segment of the Spinal Cord of Both Male and Female Mice

Our previous study using Western blot analysis of the whole spinal cord showed that phospho-p38 MAPK was significantly increased in the spinal cords of ddC-treated female mice but not in those of ddC-treated male mice, despite the fact that microglia were activated in both female and male mice (increase in CD11b and Iba1 expression, respectively) [5]. To ascertain if there were sex-specific phospho-p38 MAPK expression on the lumbar sections of the spinal cord, immunofluorescence staining of phospho-p38 MAPK was performed on the lumbar sections of the spinal cord from both sexes 7 days after ddC treatment. There was a significant increase in phospho-p38 MAPK staining intensity on the lumbar sections of the spinal cord of both females and males treated with ddC compared to treatment with the vehicle (*p* < 0.05; Figure 4C and Figure 4F, respectively). When male and female mice were compared, there were sex differences in phospho-p38 MAPK staining intensity in control and ddC-treated mice; i.e., phospho-p38 MAPK was more in female than male mice (*p* < 0.01; Figure 5).

### 2.3. Minocycline Suppressed ddC-Induced Mechanical Allodynia in Both Male and Female Mice

To determine if microglia activation contributed to the development of mechanical allodynia in ddC-induced neuropathic pain, minocycline, a microglia inhibitor, was injected 16 h before the first ddC administration and alongside with ddC for five consecutive days in both male and female mice. Treatment with ddC (25 mg/kg) significantly reduced the withdrawal threshold to mechanical stimuli of both male and female mice on day 7 after the first ddC injection compared to baseline values (male: 1.400 [1.375–1.475] g compared to 4.750 [4.300–4.825] g, *p* = 0.0022, female: 1.350 [1.200–1.475] g compared to 4.500 [3.900–4.750] g, *p* = 0.0022, respectively) and to control vehicle-treated mice (male: 1.400 [1.375–1.475] g compared to 4.400 [4.100–4.850] g, *p* = 0.0006, female: 1.350 [1.200–1.475] g compared to 4.650 [4.250– 4.850] g, *p* = 0.0001; Figure 6A and Figure 6B, respectively). Both male and female mice treated with ddC (25 mg/kg) plus minocycline (50 mg/kg) for 5 consecutive days suppressed mechanical allodynia on day 7 as they had withdrawal threshold with values (male: 2.900 [2.800–3.200] g, female: 2.600 [2.175–2.850] g) that were significantly higher than those mice treated with ddC plus the vehicle (male: 1.400 [1.375–1.475] g, *p* = 0.0062, female: 1.350 [1.200–1.475] g, *p* = 0.0009; Figure 6A and Figure 6B, respectively). However, there were no significant sex differences in the effect of minocycline on ddC-induced mechanical allodynia (*p* > 0.05; Figure 7).

### 2.4. Inhibition of p38 MAPK Prevented ddC-Induced Mechanical Allodynia in Both Male and Female Mice

In order to assess whether p38 MAPK signaling contributed to the development of ddC-induced neuropathic pain, adezmapimod (SB203580), a selective p38 MAPK inhibitor, was injected alongside ddC for five consecutive days in both male and female mice. Treatment with ddC (25 mg/kg) significantly reduced the withdrawal threshold to mechanical stimuli of both male and female mice on day 7 after the first ddC administration compared to baseline values (male: 1.400 [1.375–1.625] g compared to 4.650 [4.600–4.775] g, *p* = 0.0022, female: 1.500 [1.400–1.625] g compared to 4.450 [4.200–4.675] g, *p* = 0.0022, respectively) and to control vehicle-treated mice (male: 1.400 [1.375–1.625] compared to 4.650 [4.450–5.000] g, *p* = 0.0001, female: 1.500 [1.400–1.625] g compared to 4.450 [4.075–4.800], *p* = 0.0001; Figure 8A and Figure 8B, respectively). Mice treated with ddC (25 mg/kg) plus adezmapimod (30 mg/kg; twice) for five consecutive days had a withdrawal threshold with values (male: 4.200 [4.075–4.400] g, female: 3.600 [3.400–3.700] g) that were significantly higher than those of mice treated with ddC plus the vehicle (male: 1.400 [1.375–1.625] g, *p* = 0.0001, female: 1.500 [1.400–1.620] g, *p* = 0.0001; Figure 8A and Figure 8B, respectively). However, there were significant differences between male and female mice in the effect of adezmapimod on ddC-induced mechanical allodynia; i.e., the percent change from baseline values for male mice (4.650 [4.500–4.825] g) were reduced more than female mice (4.200 [4.100–4.700] g); *p* = 0.0047; Figure 9).

## 3. Discussion

Recently, we reported that there were sex differences in microglia activation and p38 MAPK expression in the spinal cords of male and female BALB/c mice with ddC-induced mechanical allodynia after real-time reverse transcriptase-polymerase chain reaction and Western blot analysis [1]. In this study, using immunofluorescence staining, we confirm that microglia are activated differently in male and female mice with ddC-induced neuropathic pain model. There was a significant increase in microglia markers CD11b and Iba1 staining intensity on the lumbar spine sections of ddC-treated male mice, while there was a significant upregulation in CD11b, but not Iba1, staining intensity on the lumbar spine sections of ddC-treated female mice. In addition, treatment with ddC increased phospho-p38 MAPK staining intensity on the lumbar spine sections of both male and female mice. Sex differences were found in the staining intensity of microglia markers (CD11b and Iba1) and phospho-p38 MAPK after treatment with ddC; i.e., Iba1 was higher in male than female mice, while CD11b and phospho-p38 MAPK were higher in female than in male mice treated with ddC. Intraperitoneal administration of minocycline, a microglia inhibitor, and adezmapimod, a selective p38 MAPK inhibitor, suppressed mechanical allodynia induced by ddC in both male and female mice. The effect of the p38 MAPK inhibitor was more in male mice compared to female mice.

Microglia are resident immune cells that represent around 10% of total CNS cells in the brain and the spinal cord [33]. In neuropathic pain models, microglia were believed to play an important role in the pathogenesis of mechanical allodynia, primarily in male mice, while T cells played an important role in female mice [26,34]. In a previous study, mechanical allodynia was associated with microglia activation in the spinal cords of both male and female mice in ddC-induced neuropathic pain model [7]; in males, the protein expression of Iba1 was upregulated, whereas in females, the gene and protein expression of CD11b were increased. The present study confirms that both male and female microglia are important for the pathogenesis of mechanical allodynia, as revealed by CD11b and Iba1 immunostaining in male mice and CD11b immunostaining in female mice. The activation of microglia showed sex differences whereby female mice had stronger immunofluorescence staining of CD11b compared to male mice, and male mice had stronger Iba1 staining than female mice. The significance of these differences in the expression of microglia activation markers during NRTI-induced neuropathic pain need to be explored further.

Mitogen-activated protein kinases (MAPKs) are highly conserved family of ubiquitous proline-directed, protein serine/threonine kinases [35]. There are three distinct MAPK pathways in mammalian cells, which include extracellular signal-regulated kinases (ERK1/2), p38 MAPK, and c-Jun N-terminal kinase isoforms (c-JNK) pathways. The p38 MAPK plays a critical role in inflammatory responses and microglia activation [23]. Previously, the protein expression of phospho-p38 MAPK was upregulated in ddC-treated female, but not male mice, compared to control vehicle-treated mice [7]. Here, we detected a significant increase in phospho-p38 MAPK staining intensity on the lumbar spine sections of both male and female mice treated with ddC; however, female mice had stronger phospho-p38 MAPK immunofluorescence staining compared to male mice. The reason for the discrepancy between the results of the first and the current study could be possibly due to the fact that we used the whole spinal cord to evaluate the protein expression using the automated Wes™ capillary-based protein electrophoresis (ProteinSimple, San Jose, CA, USA) in the first study [7], and the lumbar spine specifically for immunofluorescence staining in the current study. However, the fact that ddC-treated female mice had higher phospho-p38 MAPK staining intensity than ddC-treated male mice is in line with the previous findings [7]. Surprisingly, control female mice had higher phospho-p38 MAPK staining intensity than control male mice. This suggests that female mice had a higher basal p38 MAPK activity than male mice.

Minocycline is a second-generation tetracycline that is widely used as an antibiotic, and has anti-inflammatory, anti-apoptotic, and antioxidant effects [36]. Minocycline is highly lipophilic, can easily cross the blood–brain barrier (BBB), and has been shown to have neuroprotective effects in animal models of CNS pathologies [37]. Minocycline preserves the integrity of the BBB, inhibits microglia activation, and reduces the influx of immune cells to the CNS during inflammation in both animal models and humans [38,39]. The anti-inflammatory effects of minocycline are carried out through microglia inhibition, thus reducing the production of inflammatory cytokines, such as IL-1β, IL-6, and TNF-α, and inhibiting the p38 MAPK pathway. In addition, minocycline inhibits T-cell migration into the CNS [40]. The anti-inflammatory effects of minocycline have been observed in patients with chemotherapy-induced [41], diabetic [42], and leprotic neuropathies [43]. Several studies have shown that minocycline reduced mechanical allodynia in animal models through microglia inhibition. Mechanical allodynia was attenuated by the administration of minocycline in different models of neuropathic pain: ddC-induced neuropathic pain [8], antineoplastic-induced neuropathic pain [44,45], and spinal cord injury [17,46]. A preclinical study showed sex differences in reversing mechanical allodynia in a spinal nerve injury (SNI)-induced neuropathic pain model in which minocycline had anti-allodynic effects only in male mice [26]. However, minocycline also prevented allodynia effectively in female mice after spinal cord injury [47] and ddC-induced neuropathic pain [8]. The findings of the current study show that there were no sex differences in the effect of minocycline on ddC-induced mechanical allodynia in which spinal microglia were activated in both male and female mice.

Previous studies have shown that intrathecal administration of a p38 MAPK inhibitor (SB203580) suppressed mechanical allodynia induced by CCI [25] and SNI [26] in male mice, without having an effect in female mice. However, an intraperitoneal injection of p38 MAPK inhibitor suppressed mechanical allodynia in both in a sex-independent manner in the CCI model [25]. The CCI study suggests that the sex-dependent p38 activation and signaling are confined to the spinal cord [25,48]. In the current study, mechanical allodynia was prevented by the intraperitoneal p38 MAPK inhibitor (adezmapimod) in both male and female mice, but in a sex-dependent manner. The effect of adezmapimod was more in male than in female mice, possibly because male mice had less phospho-p38 MAPK in the spinal cord compared to female mice, as reported previously [7]. The differences between the previous studies and the current study are of clinical importance in various ways. Firstly, they show that neuropathic pain is heterogenous in terms of microglia activation and p38 MAPK signaling. Mechanically induced neuropathic pain by direct injury to the nerves differ from antiretroviral drug-induced neuropathic pain. The effects of the systemic administration of the p38 MAPK inhibitor were sex-independent in the CCI model, whereas they were sex-dependent in the antiretroviral drug-induced neuropathic pain model. Thus, in clinical trials, the different types of neuropathic pain and gender should be differently grouped when studying the effect of p38 MAPK inhibitors after the systemic administration of drugs (the route most used for drug administration in the management of neuropathic pain). From the previous findings and our findings, it is possible that p38 MAPK inhibitors might be less effective in females with antiretroviral drug-induced neuropathic pain compared to males, while females with mechanically induced neuropathic pain (such as spinal cord injury or nerve compression) as well as males from both groups might respond better. This important information could be lost if different types of neuropathic pain are pooled together, as happens in some clinical trials.

There are limitations that exist in directly reflecting sex differences about neuropathic pain obtained from animal studies in clinical studies. Recent studies suggest that the estrous cycle has minimal effects on behavioral studies in rodents, including pain [49,50], while the phase of the menstrual cycle has an effect in women, and hormonal levels may affect the severity of chronic pain [51,52]. Secondly, preclinical studies evaluated mostly provoked pain while humans experience and express the quality of their pain to both provoked and spontaneous pain, which affects the effectiveness of drugs as the pain measured in humans is multifaceted. More specific to microglia, there are important differences between human and rodent microglia; for example, toll- like receptor 4 (TLR4) is highly expressed in rodent microglia and slightly expressed in human microglia [53]. These differences might also be reflected in the response to microglia inhibitors, such as minocycline, which has been found to be effective in various models of neuropathic pain but did not produce clinically significant benefits for patients in clinical trials of neuropathic pain [54,55]. Sometimes, the timing of administration also influences the responses to microglia inhibitors. For example, minocycline alone was effective in preventing chemotherapy-induced neuropathic pain (CINP) when administered prophylactically [18] but could not alleviate established CINP in rodents [56]. Thus, more robust multistep studies are necessary before trying to translate preclinical studies conducted on rodents to human beings. These include having models that closely mimic the type of neuropathic pain being studied, studying various timings of drug administration and drug pharmacokinetics, as well as utilizing cell or tissue models of human origin to complement the studies conducted on rodents.

## 4. Materials and Methods

### 4.1. Animals

A total of 90 age-matched male and female BALB/c mice (8–12 weeks old; 20–25 g; *n* = 45 for each sex) were provided by the Animal Resource Center at the Health Sciences Center (HSC), Kuwait University, Kuwait. Animal care was in accordance with the guidelines established by the European Parliament and of the Council (Directive 2010/63/EU). All procedures used in the study were approved by the Ethical Committee for the Use of Laboratory Animals in Teaching and in Research, HSC (Ref: 23/VDR/EC/, Date 5/5/2021). All animals were housed in polycarbonate cages and maintained on a 12 h light/dark cycle (lights on at 06:00 a.m. and lights off at 18:00 p.m.) under set temperature (23 ± 1 °C), with ad libitum access to food and water. All procedures were performed at the same period (11:00 a.m. and 19:00 p.m.) to exclude diurnal variations in response. Experimental mice were divided into groups, as shown in Table 1.

### 4.2. Drug Preparation and Administration

2′,3′-dideoxycytidine (ddC) (Sigma-Aldrich, St. Louis, MO, USA) was freshly prepared in normal saline and administered intraperitoneally (i.p.) to mice at a dose of 25 mg/kg in a volume of 10 mL/kg for five consecutive days, as previously described [7].

Minocycline (Sigma-Aldrich, St. Louis, MO, USA) was dissolved in phosphate-buffered saline (PBS) and administered i.p. at a dose of 50 mg/kg, similarly to what was reported previously [8]. Minocycline (50 mg/kg) was injected 16 h before the first ddC injection. After that, it was given once daily for five consecutive days one hour before ddC administration (see Figure 10).

Adezmapimod (SB203580) (Selleckchem, Houston, TX, USA) was dissolved in normal saline in a glass vial covered with aluminum foil followed by ultra-sonication to dissolve completely and administered i.p. twice daily (8 h between) at a dose of 30 mg/kg across 5 days (see Figure 10), similar to what was reported previously [24]. All drugs were freshly prepared on the day of the experiment.

In this study, the effects of the drugs were compared to that of their vehicles.

### 4.3. Assessment of Mechanical Allodynia

Mechanical allodynia was measured using the dynamic plantar aesthesiometer (DPA) (Ugo Basile, Gemonio, Italy), as previously reported [7]. Briefly, all mice were left to acclimatize for about 90 min inside a plastic enclosure on top of a perforated platform every day for three days before the experiment. On the day of the experiment, mice were left for 60 min to habituate to the aesthesiometer. The DPA’s system was programmed to automatically uplift a touch stimulator unit that was directly placed beneath the hind paw as the force increased linearly (0.25 g/s). The force either at the time of paw withdrawal in response to the mechanical stimuli or at a cut-off force of 5 g was detected and automatically recorded by the DPA. At least four different readings were taken for each mouse. The baseline readings were assessed before minocycline or adezmapimod treatment and before inducing neuropathic pain by ddC.

### 4.4. Tissue Fixation, Processing, and Paraffine Embedding

Mice were anesthetized with halothane and euthanized by decapitation on day 7 after the first ddC administration. The spinal cords were extracted as previously described [57] and fixed by 4% paraformaldehyde (PFA) in 1× PBS (pH 7.4) at 4 °C for 2 h. Then, the spinal cords were transferred to 25% sucrose at 4 °C overnight. The tissues were stored in 25% sucrose with a few droplets of 10% sodium azide [57]. On the second day, the spinal cords were trimmed using a dissecting blade to isolate the lumbar (L1–L6) enlargement, placed in cassettes, and processed using an automated tissue processor overnight (ATP 1000 Automatic Tissue Processor). The next day, the tissues were embedded in paraffin (Jung Histoembedder, Leica, Wetzlar, Germany) according to standard protocols. The paraffin blocks were sectioned coronally (5 μm thickness) using microtome machine (Jung Histocut, Leica) and mounted on bond plus slides (Leica, Wetzlar, Germany) for immunofluorescence. 

### 4.5. Immunofluorescence

Coronal sections obtained from each lumbar segment of the spinal cord were deparaffinized in xylene and rehydrated through descending grades of alcohol. Antigen retrieval was performed by immersing the slides in a 10 mM sodium citrate buffer (2.94 g of dihydrate tri-sodium citrate in 1000 mL of distilled water, pH 6.0) on a hot plate (boiling) for 10 min. After that, the sections were washed with PBS (2× each 8 min) followed by 8 min wash with PBS solution containing 0.1% triton (Sigma Aldrich, St. Louis, MO, USA). The non-specific sites were blocked for 50 min at room temperature using PBS solution containing 1% bovine serum albumin (BSA) in a humidified chamber. The spinal cord lumbar sections were incubated with primary rabbit anti-CD11b (1:100; Abcam, Cambridge, UK), rabbit anti-Iba1 (1:2000; Boster Bio, Pleasanton, CA, USA), and rabbit anti-phospho-p38 MAPK (1:50; Cell Signaling, Danvers, MA, USA) overnight at room temperature in a humidified chamber. On the following day, these sections were washed with PBS (3× each 8 min) and incubated with the secondary antibody (1: 2000; donkey anti-rabbit IgG [Alexa Fluor 488]; Life Technologies, Carlsbad, CA, USA) for 2 h at room temperature in a dark humidified chamber. All the antibodies were diluted in PBS containing 0.1% Triton and 1% BSA. After the incubation period, the sections were washed again with PBS (3× each 8 min). Finally, the slides were mounted with DPX mounting media (Sigma-Aldrich, St. Louis, MO, USA) and covered with cover slips overnight. On the next day, the slides were observed under confocal microscopy (Zeiss LSM 700 META, Carl Zeiss microscopy, New York, NY, USA), and fluorescence intensity of each section was calculated. Images of these sections were taken at 20× objective. Three images from each section were taken and analyzed. The images were presented using Adobe Photoshop (version 23.1).

### 4.6. Statistical Analysis

Data were analyzed using GraphPad Prism software (version 9.00; GraphPad Software Inc., San Diego, CA, USA). Immunofluorescence intensity data were statistically analyzed using Mann–Whitney U test to assess the differences between vehicle and ddC groups, and two-way analysis of variance (ANOVA) followed Bonferroni’s multiple comparisons test to assess differences between the sexes. Two-way repeated measures ANOVA followed by Tukey’s multiple comparisons test were used to compare the withdrawal threshold between ddC plus vehicle-treated, ddC plus minocycline-treated, ddC plus adezmapimod-treated, and control-vehicle treated mice at baseline and day 7. Mann–Whitney U test was used to compare the percent change of withdrawal threshold from baseline after drug treatment between male and female mice. The differences were considered significant when *p* < 0.05. The results were presented as median and interquartile range. 

## 5. Conclusions

In conclusion, the current study confirms that there are sex-dependent differences in the expression of microglia activation markers and phospho-p38 MAPK in the lumbar segment of the spinal cord of BALB/c mice during ddC-induced mechanical allodynia. Minocycline, an inhibitor of microglia activation, and adezmapimod, a p38 MAPK inhibitor suppressed mechanical allodynia that had been induced by ddC in both male and female mice. Therefore, the inhibition of microglia activation and p38 MAPK signaling could be useful for managing antiretroviral drug-induced neuropathic pain in both sexes, while male patients could respond better to a p38 MAPK inhibitor than female patients. This new information could be important for the type of neuropathic pain-specific drug development and personalized treatment, taking into consideration other important differences, such as sex. Our findings could help in the search of therapeutic drugs for neuropathic pain caused by neurological diseases in humans, as clinical studies have revealed the effects of minocycline to treat CNS pathologies [40].

## Figures and Tables

**Figure 1 ijms-24-12805-f001:**
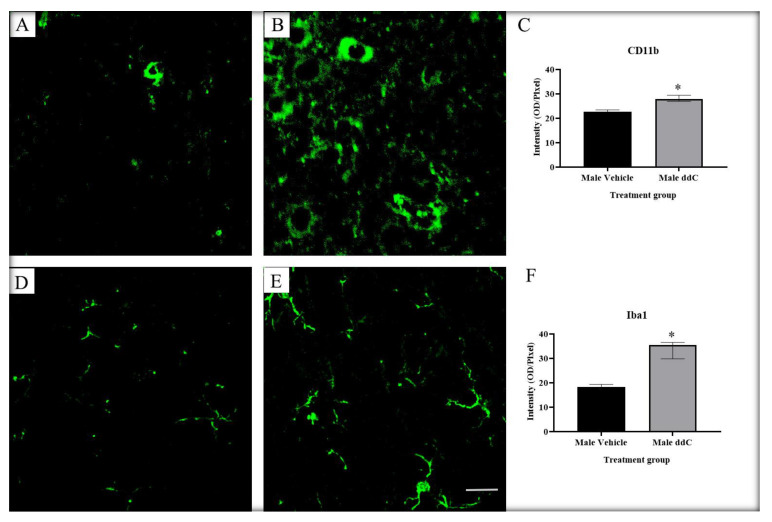
Immunofluorescent images of microglia from the lumbar sections of the spinal cords immunostained using CD11b and Iba1 on day 7 post-ddC administration of male BALB/c mice. Immunofluorescence staining of CD11b in (**A**) control vehicle-treated and (**B**) ddC-treated male mice. (**C**) Average immunofluorescence staining of CD11b in male mice. Immunofluorescence staining of Iba1 in (**D**) control vehicle-treated and (**E**) ddC-treated male mice. (**F**) Average immunofluorescence staining of Iba1 in male mice. (**C**,**F**) Each bar represents the median and interquartile range obtained from four mice. * *p* < 0.05 compared to control vehicle-treated male mice (Mann–Whitney U test). (**A**,**B**,**D**,**E**) Scale bar: 20 μm.

**Figure 2 ijms-24-12805-f002:**
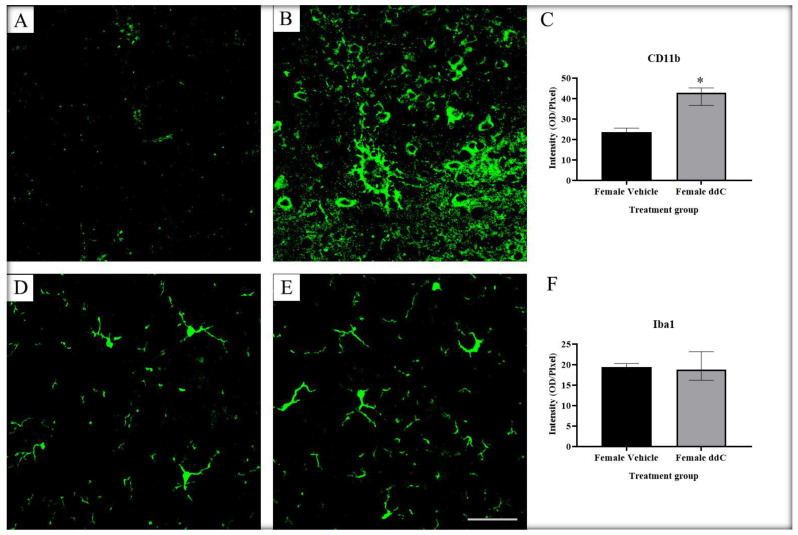
Immunofluorescent images of microglia from the lumbar sections of the spinal cords immunostained using CD11b and Iba1 on day 7 post-ddC administration of female BALB/c mice. Immunofluorescence staining of CD11b in (**A**) control vehicle-treated and (**B**) ddC-treated female mice. (**C**) Average immunofluorescence staining of CD11b in female mice. Immunofluorescence staining of Iba1 in (**D**) control vehicle-treated and (**E**) ddC-treated female mice. (**F**) Average immunofluorescence staining of Iba1 in female mice. (**C**,**F**) Each bar represents the median and interquartile range obtained from four mice. * *p* < 0.05 compared to control vehicle-treated female mice (Mann–Whitney U test). (**A**,**B**,**D**,**E**) Scales bar: 50 μm.

**Figure 3 ijms-24-12805-f003:**
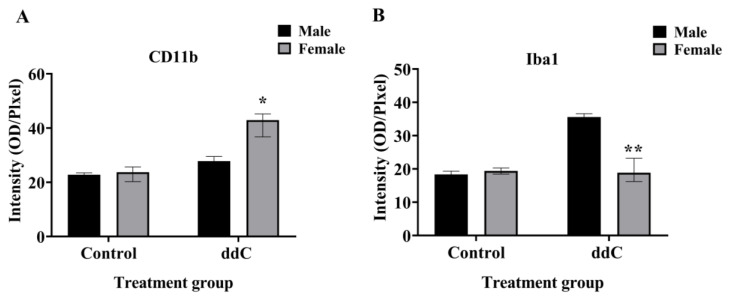
Sex differences in microglia markers (**A**) CD11b and (**B**) Iba1 immunostaining intensity on the lumbar sections of the spinal cords of male and female BALB/c mice on day 7 after ddC administration. Each bar represents the median and interquartile range obtained from four mice. * *p* < 0.05, ** *p* < 0.01 compared to ddC-treated male mice (two-way ANOVA followed by Tukey’s multiple comparisons test).

**Figure 4 ijms-24-12805-f004:**
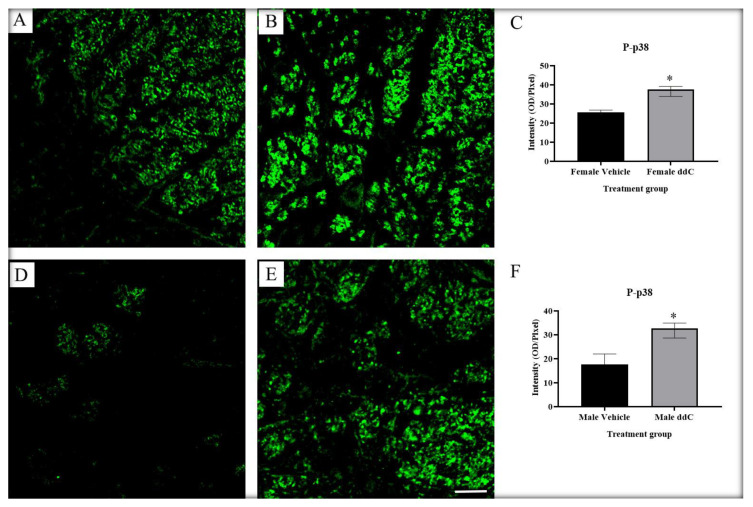
Immunofluorescence staining of phosphorylated p38 MAPK (P-p38) on the lumbar sections of the spinal cords on day 7 post-ddC administration of female and male BALB/c mice. Immunofluorescence staining of P-p38 in (**A**) control vehicle-treated and (**B**) ddC-treated female mice. (**C**) Average immunofluorescence staining of P-p38 in female mice. Immunofluorescence staining of P-p38 in (**D**) control vehicle-treated and (**E**) ddC-treated male mice. (**F**) Average immunofluorescence staining of P-p38 in male mice. (**C**,**F**) Each bar represents the median and interquartile range obtained from four mice. * *p* < 0.05 compared to control vehicle-treated mice (Mann–Whitney U test). (**A**,**B**,**D**,**E**) Scales bar: 20 μm.

**Figure 5 ijms-24-12805-f005:**
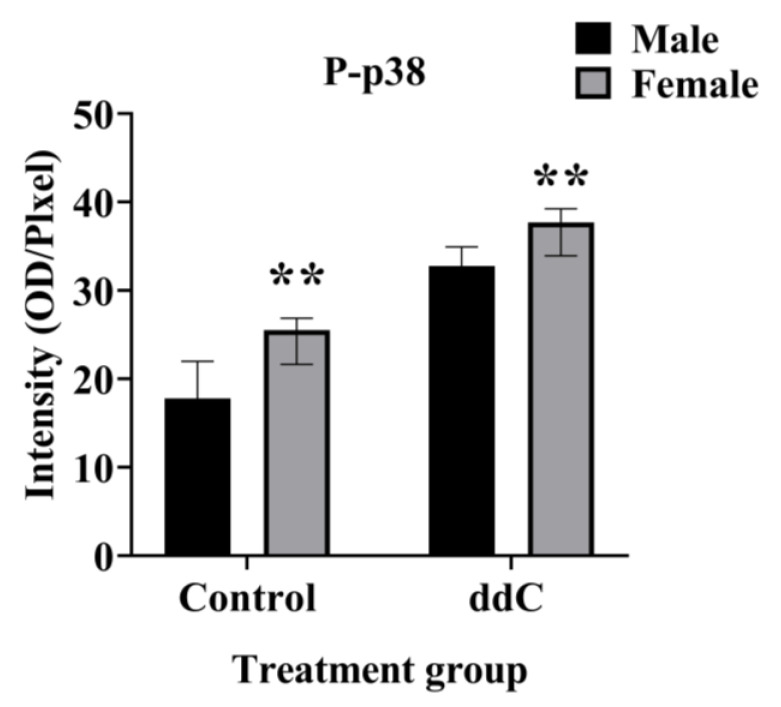
Sex differences in phosphorylated p38 MAPK (P-p38) staining intensity on the lumbar sections of the spinal cord on day 7 after ddC administration. Each bar represents the median and interquartile range obtained from four mice. ** *p* < 0.01 compared to ddC-treated male mice (two-way ANOVA followed by Tukey’s multiple comparisons test).

**Figure 6 ijms-24-12805-f006:**
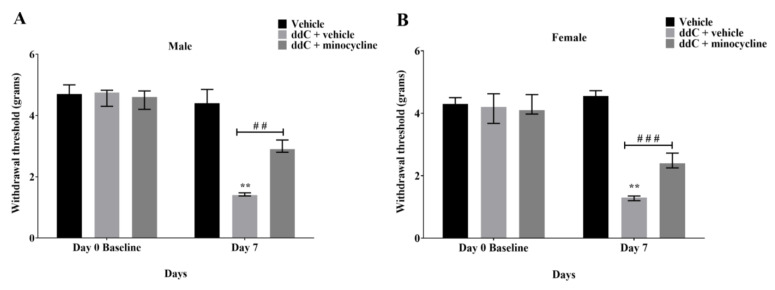
Minocycline suppressed the development of mechanical allodynia induced by ddC in both male and female BALB/c mice. (**A**) Withdrawal threshold to mechanical stimuli of male mice before (baseline) and on day 7 post-ddC (25 mg/kg) administration and treatment with minocycline (50 mg/kg). (**B**) Withdrawal threshold to mechanical stimuli of female mice before (baseline) and on day 7 post-ddC (25 mg/kg) administration and treatment with minocycline (50 mg/kg). Each bar represents the median and interquartile range obtained from six to seven mice. ** *p* < 0.01 compared to baseline values for ddC-treated mice (Mann–Whitney U test). ## *p* < 0.01 and ### *p* < 0.001 compared to mice treated with ddC plus vehicle at day 7 (two-way repeated measures ANOVA followed by Tukey’s multiple comparisons test).

**Figure 7 ijms-24-12805-f007:**
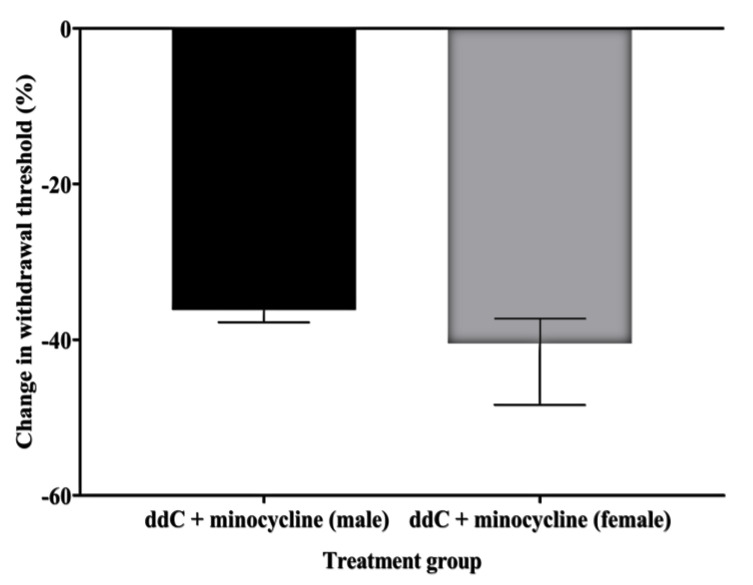
Change in withdrawal threshold (%) from baseline values to mechanical stimuli on day 7 for ddC plus minocycline-treated male and female BALB/c mice. Each bar represents the median and interquartile range obtained from six to seven mice. No significant differences between the groups were detected (Mann–Whitney U test).

**Figure 8 ijms-24-12805-f008:**
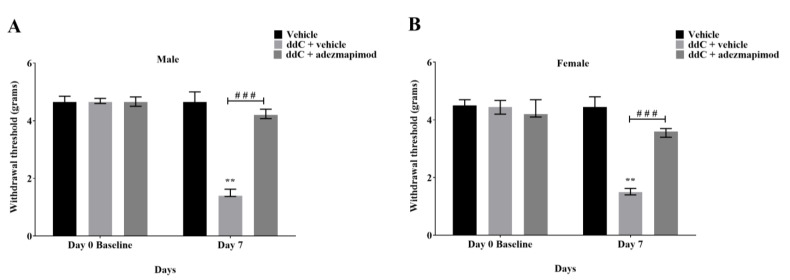
Adezmapimod suppressed the development of mechanical allodynia induced by ddC in both male and female BALB/c mice. (**A**) Withdrawal threshold to mechanical stimuli of male mice before (baseline) and on day 7 post-ddC (25 mg/kg) administration and treatment with adezmapimod (30 mg/kg). (**B**) Withdrawal threshold to mechanical stimuli of female mice before (baseline) and on day 7 post-ddC (25 mg/kg) administration and treatment with adezmapimod (30 mg/kg). Each bar represents the median and interquartile range obtained from six to seven mice. ** *p* < 0.01 compared to baseline values for ddC-treated mice (Mann–Whitney U test). ### *p* < 0.001 compared to mice treated with ddC plus vehicle on day 7 (two-way repeated measures ANOVA followed by Tukey’s multiple comparisons test).

**Figure 9 ijms-24-12805-f009:**
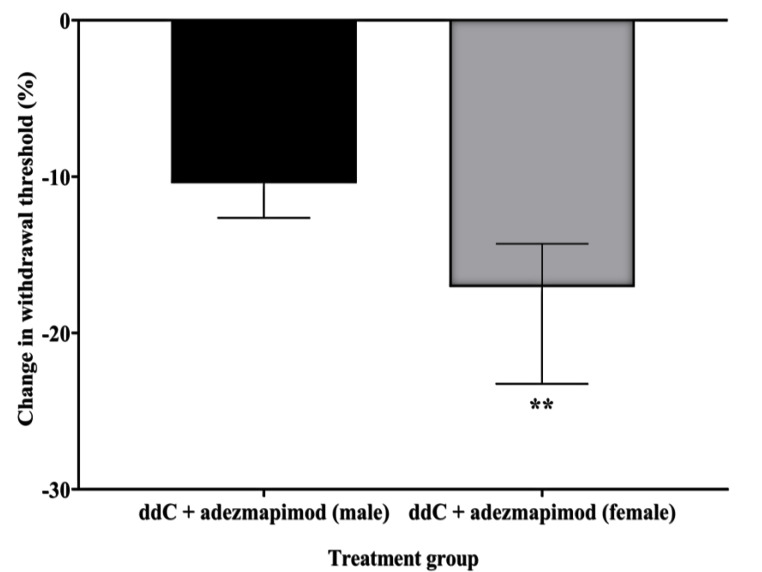
Change in withdrawal threshold (%) from baseline values to mechanical stimuli on day 7 for ddC plus adezmapimod-treated male and female BALB/c mice. Each bar represents the median and interquartile range obtained from six to seven mice. ** (*p* < 0.01) statistically significant differences between the groups were detected (Mann–Whitney U test).

**Figure 10 ijms-24-12805-f010:**
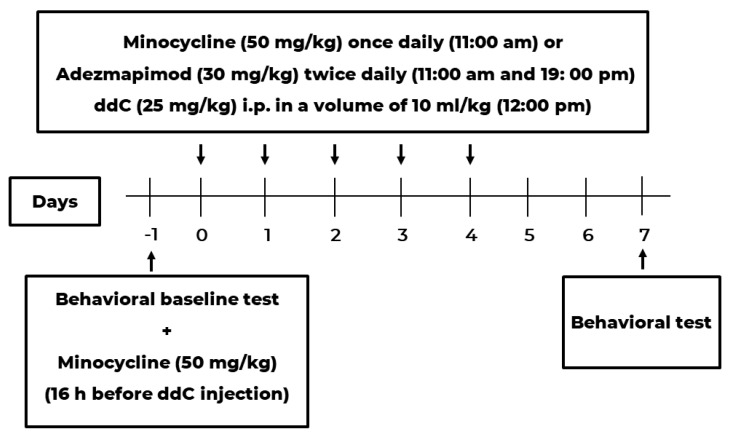
Drug administration schedule for treatment with minocycline and adezmapimod to prevent ddC-induced mechanical allodynia.

**Table 1 ijms-24-12805-t001:** The allocation of male and female BALB/c mice in the experimental groups.

Experimental Group	Number of Male Mice	Number of Female Mice
vehicle only *	16	16
ddC *	16	16
ddC + minocycline	7	6
ddC + adezmapimod	6	7

* The sixteen mice for each sex (for the vehicle only and ddC group) are a total of three separate groups of experiments; i.e., four for immunofluorescence, six for the minocycline experiment, and six for the adezmapimod experiment.

## Data Availability

Data will be made available on request.

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
