# Peer review of "Microglia and p38 MAPK Inhibitors Suppress Development of Mechanical Allodynia in Both Sexes in a Mouse Model of Antiretroviral-Induced Neuropathic Pain"

_ijms, 2023, doi:10.3390/ijms241612805_

Round 1

Reviewer 1 Report

Dear. Author,

This manuscript is well organized and has no major flaws in its logical development. Furthermore, the conclusions drawn from this study provide useful and analytical information about antiretroviral-induced neuropathic pain.

However, although microglia and p38 MAPK inhibitors inhibit mechanical allodynia in both sexes, there seems to be a difference between the sexes. This needs to be further explained. I would also recommend adding the authors' thoughts on the clinical application of the findings.

Thank you.

Author Response

This manuscript is well organized and has no major flaws in its logical development. Furthermore, the conclusions drawn from this study provide useful and analytical information about antiretroviral-induced neuropathic pain. However, although microglia and p38 MAPK inhibitors inhibit mechanical allodynia in both sexes, there seems to be a difference between the sexes. This needs to be further explained.

Answer: There were sex differences  in response to treatment with the p38 MAPK inhibitor (adezmapimod) and the possible reason was mentioned in the discussion section. “However, the effect of adezmapimod was more in male than in female mice, possibly because the male mice had less phospho-p38 MAPK in the spinal cord compared to female mice as reported previously (Alhadlaq and Masocha, 2023)”   However, we didn’t observe sex differences in response to treatment with minocycline.

I would also recommend adding the authors' thoughts on the clinical application of the findings.

Answer: Thank you very much. The clinical application of the findings has been stated in the conclusion section.

Reviewer 2 Report

The total number of animals used for the experiments, the allocation of animals to experimental groups, and the number of animals in each group, clear description of positive and negative control groups should be included in Matherials and Methods section.

Minor editing is required

Author Response

The total number of animals used for the experiments, the allocation of animals to experimental groups, and the number of animals in each group, clear description of positive and negative control groups should be included in Materials and Methods section.

Answer: The total number of animals used for the experiments has been specified in the methods section (a total of 90 mice i.e., 45 for each sex).

The allocation of animals to experimental groups and the number of animals in each group are specified in a table.

Experimental group

Number of male mice

Number of female mice

Vehicle only*

16

16

ddC *

16

16

ddC+ minocycline

7

6

ddC + adezmapimod

6

7

*The sixteen mice for each sex (for the vehicle only and ddC group) are a total of three separate groups of experiments i.e., four for immunofluorescence, six for the minocycline experiment and six for the adezmapimod experiment.

Vehicle (normal saline or PBS) was used as a negative control to the drugs.  One of the drugs minocycline has already been used in our lab previously but only  on ddC-treated female mice (Aly et al., 2019).

Reviewer 3 Report

The Authors present the results that showed many positive effects of P38 and MAPK inhibitors on neuropathic pain in both sexes in a mouse model of antiretroviral-induced neuropathic Pain. Administration of minocycline, an inhibitor of microglia activation, and adezmapimod, a selective p38 MAPK inhibitor, suppressed mechanical allodynia in both sexes at day 7 after neuropathic pain establishment.

The topic is interesting and seems to be important bearing in mind the need to treat many patients with neuropathic complications with limited treatment options. The experiments were designed and conducted via interesting ways and performed in accordance with a dedicated method.

1.     My main question to the authors is whether the effects that had been studied are also observed in patients treated with Minocycline and p38 MAPK inhibitor drugs available in the market? Minocycline is an antibacterial drug that was introduced to the market several years ago and, I think, there are already many patients treated with this drug or in combination with p38 MAPK inhibitor drugs.

2.     I am also wondering about the effect of Minocycline and p38 MAPK inhibitors on the central nerve system. Though minocycline is originally developed as an antibacterial drug, there are several reports on its neuroprotection and its high efficacy in penetrating the blood brain barrier. Please discuss the effects of Minocycline on the CNS in the discussion.

3.     Evidence of the pharmacological effects on the brain and BBB permeability of antibacterial drugs like minocycline is increasing gradually, but not all of these effects have been revealed. These issues should be reflected in the discussion of the presented results.

4.     The authors have mentioned that p38 MAPK was upregulated significantly only in female mice in the introduction section. Will p38 MAPK inhibitors be more effective in treating pain in female patients? Please discuss it in the discussion if there are any supporting reports on this finding.

5.     When there are several models of neuropathic pain like Partial sciatic nerve transection, Chronic constriction injury and Sciatic nerve compression or ligation why did the authors choose to induce neuropathic pain using 2¢-3¢-dideox- 12 ycytidine (ddC), a nucleoside reverse transcriptase inhibitor (NRTI). Is there any specific reason why the authors choose this chemical over other well-established models of neuropathic pain? Do the authors believe the same mechanism will be involved in other models of neuropathic pain? And will the efficacy of minocycline and p38 MAPK inhibitors remain the same in other models of neuropathic pain. Please briefly discuss in the introduction.

Minor:

Question: I don't know how many animals were used in each experimental group.

Minor editing of English language required

Author Response

The Authors present the results that showed many positive effects of P38 and MAPK inhibitors on neuropathic pain in both sexes in a mouse model of antiretroviral-induced neuropathic Pain. Administration of minocycline, an inhibitor of microglia activation, and adezmapimod, a selective p38 MAPK inhibitor, suppressed mechanical allodynia in both sexes at day 7 after neuropathic pain establishment.

The topic is interesting and seems to be important bearing in mind the need to treat many patients with neuropathic complications with limited treatment options. The experiments were designed and conducted via interesting ways and performed in accordance with a dedicated method.

Answer: Thank you very much.

  1. My main question to the authors is whether the effects that had been studied are also observed in patients treated with Minocycline and p38 MAPK inhibitor drugs available in the market? Minocycline is an antibacterial drug that was introduced to the market several years ago and, I think, there are already many patients treated with this drug or in combination with p38 MAPK inhibitor drugs.

Answer: Evidence on the anti-inflammatory effects of minocycline in patients with chemotherapy-induced, diabetic, and leprotic neuropathies were specified in the discussion section. On the other hand, the effects of p38 MAPK inhibitor on pain remain uncertain, yet some studies showed improvement in immune molecules during inflammation ( Sarov-Blat et al., 2010; Goldstein, et al., 2010).   

  1. I am also wondering about the effect of Minocycline and p38 MAPK inhibitors on the central nerve system. Though minocycline is originally developed as an antibacterial drug, there are several reports on its neuroprotection and its high efficacy in penetrating the blood brain barrier. Please discuss the effects of Minocycline on the CNS in the discussion.

      Answer: The effects of Minocycline on the CNS has now been stated in the discussion section.

  1. Evidence of the pharmacological effects on the brain and BBB permeability of antibacterial drugs like minocycline is increasing gradually, but not all of these effects have been revealed. These issues should be reflected in the discussion of the presented results.

Answer: Minocycline preserves the integrity of the BBB, inhibits microglia activation and reduces the influx of immune cells to the CNS during inflammation in both animal models and  humans (Strickland et al., 2021; Lu et al., 2022). This aspect has now been added to the discussion section.

  1. The authors have mentioned that p38 MAPK was upregulated significantly only in female mice in the introduction section. Will p38 MAPK inhibitors be more effective in treating pain in female patients? Please discuss it in the discussion if there are any supporting reports on this finding.

Answer: From our findings basal p38 MAPK was higher in female mice than in male mice. Moreover, there was higher p38 MAPK in female mice than in male mice after treatment with ddC.  Male mice were more responsive to the p38 MAPK inhibitor than female mice most likely because the male mice had less p38 MAPK to inhibit than the female mice. Thus it is possible that male patients might be more responsive to p38 MAPK inhibition than female patients, although the drug might work in both sexes.

  1. When there are several models of neuropathic pain like Partial sciatic nerve transection, Chronic constriction injury and Sciatic nerve compression or ligation why did the authors choose to induce neuropathic pain using 2¢-3¢-dideox- 12 ycytidine (ddC), a nucleoside reverse transcriptase inhibitor (NRTI). Is there any specific reason why the authors choose this chemical over other well-established models of neuropathic pain?

Answer: In this study, ddC was used as a model NRTI-induced neuropathic pain because the recommended initial antiretroviral therapy regimens for naïve patients with HIV involves combination of drugs. NRTIs are considered cornerstones that are used in all the combination therapy used to treat HIV patients. HIV-associated neuropathic pain including that caused by antiretroviral therapy affects millions of people living with AIDS and is difficult to treat. The response to treatment differs between different types of neuropathic pain. The other models of neuropathic pain such as partial sciatic nerve transection, chronic constriction injury and sciatic nerve compression or ligation, are good models but do not serve as models for the pain caused by antiretroviral drugs.

Do the authors believe the same mechanism will be involved in other models of neuropathic pain?

Answer: The results of our previous study (Alhadlaq and Masocha, 2023) showed that there were some differences regarding the gene and protein expression of some markers in comparison with other models of neuropathic pain (Sorge et al., 2015; Dance, 2019; Taves et al., 2016). 

Will the efficacy of minocycline and p38 MAPK inhibitors remain the same in other models of neuropathic pain. Please briefly discuss in the introduction.

Answer: The efficacy of minocycline and p38 MAPK inhibitors could be different in other models of neuropathic pain. For example in a CCI model minocycline was effective alleviating allodynia in male mice but not in female mice, while it prevented the development of mechanical allodynia in the ddC model of neuropathic pain. This has now been mentioned in the introduction section.

Minor:

Question: I don't know how many animals were used in each experimental group.

Answer: The number of animals used in each experimental group has been specified in a table form and is also included in the figure legends.

Experimental group

Number of male mice

Number of female mice

Vehicle only*

16

16

ddC *

16

16

ddC+ minocycline

7

6

ddC + adezmapimod

6

7

*The sixteen mice for each sex (for the vehicle only and ddC group) are a total of three separate groups of experiments i.e., four for immunofluorescence, six for the minocycline experiment and six for the adezmapimod experiment.

Reviewer 4 Report

This manuscript has the following novelty issues. It is questionable what implications can be emphasized compared to previous studies. It would be reconsidered for re-review if the three issues below will be addressed by emphasizing the novelty of the research.

1. The development of mechanical allodynia by 2′,3′-dideoxycytidine is not a new study area for neuropathic pain.

2. The involvement of microglia in mechanical allodynia is a well-known theory, and the pain attenuation by minocycline, a microglial deactivator, is also well-established.

3. Although the analgesic effect of mechanical allodynia with adezmapimod is a not well-known area, considering that the signal transduction system of P38 for mediating the mechanical allodynia is well known, it is in the same context with many studies such as investigating the analgesic effect using SB203580 or SD-282, another type of P38 blocker.

I cannot feel serious issue on the quality of English.

Author Response

This manuscript has the following novelty issues. It is questionable what implications can be emphasized compared to previous studies. It would be reconsidered for re-review if the three issues below will be addressed by emphasizing the novelty of the research.

  1. The development of mechanical allodynia by 2′,3′-dideoxycytidine is not a new study area for neuropathic pain.

Answer: That’s true that 2′,3′-dideoxycytidine is not a new study area for neuropathic pain. However, it is an important model for antiretroviral drug  neuropathic pain and for search of new therapeutic targets for a type of neuropathic pain that affects millions of people and does not have satisfactory treatment. The sex differences is an understudied area in that type of neuropathic pain, where the majority of the studies are done on male mice.

  1. The involvement of microglia in mechanical allodynia is a well-known theory, and the pain attenuation by minocycline, a microglial deactivator, is also well-established.

Answer: That is also true and our group has published on that topic before also. However, this study sheds new light on that topic. In CCI model minocycline works in male mice but not female mice, while in this model it works in both sexes. This shows the differences in different types of neuropathic pain in terms of sex differences and response to drugs. This also reflects what happens in the clinics where one type of neuropathic pain responds well to drugs and the other type of neuropathic pain does not.

  1. Although the analgesic effect of mechanical allodynia with adezmapimod is a not well-known area, considering that the signal transduction system of P38 for mediating the mechanical allodynia is well known, it is in the same context with many studies such as investigating the analgesic effect using SB203580 or SD-282, another type of P38 blocker.

Answer: This is also true. However, the study of p38 MAPK inhibitors on antiretroviral drug-induced neuropathic pain looking at sex differences has not been done before. This new information is important for type of neuropathic pain specific drug development and personalized treatment taking into consideration other important differences such as sex.

Round 2

Reviewer 4 Report

Although the authors strongly emphasize the differences in neuropathic pain induced by gender differences, these have also been reported in many previous studies (eg. Brain Behav Immun. 2016 Jul; 55: 70–81; J Dent Res . 2016 Sep;95(10):1124-31; etc.). So, this reviewer still does not consider the novelty issue has been overcome. But, it will be acceptable if a section will be added somewhere that discusses the following parts.

1. It should be discussed what any better clinical implications have been found as compared to the previous studies.

2. It should be discussed what limitations exist in directly reflecting gender differences about neuropathic pain obtained from animal studies in clinical (eg. estrous cycle, and so on).

There is no critial issue.

Author Response

Although the authors strongly emphasize the differences in neuropathic pain induced by gender differences, these have also been reported in many previous studies (eg. Brain Behav Immun. 2016 Jul; 55: 70–81; J Dent Res . 2016 Sep;95(10):1124-31; etc.). So, this reviewer still does not consider the novelty issue has been overcome. But, it will be acceptable if a section will be added somewhere that discusses the following parts.

  1. It should be discussed what any better clinical implications have been found as compared to the previous studies.

Answer: We thank the reviewer for this suggestion. The clinical implications have been discussed in the discussion section. “Previous studies have shown that intrathecal administration of p38 MAPK inhibitor (SB203580) suppressed mechanical allodynia induced by chronic construction injury (CCI) (Taves et al., 2016) and SNI (Sorge et al., 2016) in male mice, without having an effect in female mice. However, intraperitoneal injection of p38 MAPK inhibitor suppressed mechanical allodynia in both in a sex-independent manner in the CCI model (Taves et al., 2016). The CCI study suggest that the sex-dependent p38 activation and signaling are confined to the spinal cord (Berta et al., 2016). In the current study, mechanical allodynia was prevented by intraperitoneal the p38 MAPK inhibitor (adezmapimod) in both male and female mice, but in a sex-dependent manner. The effect of adezmapimod was more in male than in female mice, possibly because the male mice had less phos-pho-p38 MAPK in the spinal cord compared to female mice as reported previously (Alhadlaq and Masocha, 2023).  The differences of the previous studies and the current study are of clinical importance in various ways. Firstly, they show that neuropathic pain is heterogenous in terms of microglia activation and p38 MAPK signalling. Mechanically induced neuropathic pain by direct injury to the nerves differ from antiretroviral drug-induced neuropathic pain. The effects of systemic administration of the p38 MAPK inhibitor was sex-independent in the CCI model while it was sex-dependent in the antiretroviral drug induced neuropathic pain model. Thus, in clinical trials the different types of neuropathic pain and gender should be differently grouped when studying the effect of p38 MAPK inhibitors after systemic administration of drugs (the route most used for drug administration in management of neuropathic pain). From the previous findings and our findings, it is possible that p38 MAPK inhibitors might be less effective in females with antiretroviral drug-induced neuropathic pain compared to males, while females with mechanically induced neuropathic pain (such as spinal cord injury or nerve compression) pain as well as males from both groups might respond better. This important information could be lost if different types of neuropathic pain are pooled together as what happens in some clinical trials.”

  1. It should be discussed what limitations exist in directly reflecting gender differences about neuropathic pain obtained from animal studies in clinical (eg. estrous cycle, and so on).

Answer: There are limitations that exist in directly reflecting gender differences about neuropathic pain obtained from animal studies in clinical studies. Recent studies suggest that the estrous cycle have minimal effects on behavioural studies in rodents including pain (Beery, 2018; Zeng et al., 2023), while the phase of the menstrual cycle has an effect in women and hormonal levels may affect the severity of chronic pain (Stening et al., 2007; Hassan et al., 2014). Secondly, preclinical studies evaluate mostly provoked pain while humans experience and express the quality of their pain to both provoked and spontaneous pain, which affects the effectiveness of drugs as the pain measured in humans is multifaceted. More specific to microglia, there are important differences between human and rodent microglia for example TLR4 is highly expressed in rodent microglia and lowly expressed in human microglia (Smith and Dragnuw, 2014). These differences might also be reflected in the response to microglia inhibitors such as minocycline, which has been found effective in various models of neuropathic pain but did not produce clinically significant benefits patients in clinical trials of neuropathic pain (Vaneldren et al., 2014, Sumitani et al., 2016). Sometimes the timing of administration also influences the responses to microglia inhibitors. For example, minocycline alone was effective in preventing chemotherapy-induced neuropathic pain when administered prophylactically (Masocha, 2014) but could not alleviate established CINP in rodents (Parvathy and Masocha, 2015). Thus, more robust multistep studies are necessary before trying to translate preclinical studies done in rodents to human beings. These include having models that closely mimic the type of neuropathic pain being studied, studying various timings of drug administration and drug pharmacokinetics as well as utilising cell or tissue models of human origin to complement the studies done in rodents.

Round 3

Reviewer 4 Report

All concerns have been well addressed. 

No problem.